# The Interactive Effect of COVID-19 Risk and Hospital Measures on Turnover Intentions of Healthcare Workers: A Time-Lagged Study

**DOI:** 10.3390/ijerph182010705

**Published:** 2021-10-12

**Authors:** Mehwish Majeed, Muhammad Irshad, Jos Bartels

**Affiliations:** 1Faculty of Management Sciences, International Islamic University, Islamabad 44000, Pakistan; mehwish.majeed@iiu.edu.pk; 2Faculty of Management Sciences, National University of Modern Languages, Islamabad 44000, Pakistan; muhammad.irshad@numl.edu.pk; 3School of Communication and Film, Department of Communication Studies, Hong Kong Baptist University, Hong Kong, China

**Keywords:** risk of COVID-19, fear of COVID-19, turnover intention, hospital measures against COVID-19, healthcare workers

## Abstract

COVID-19 has led to a global health emergency worldwide. As a result, healthcare workers undergo distress mainly due to the perceived risk of contracting the virus. Such stress might cause them to leave their jobs. In this context, the current study: (1) introduced the concept of perceived risk of COVID-19 and measured it by adapting and validating an existing scale available on the risk of infectious diseases and (2) investigated its outcomes, underlying mechanisms, and boundary conditions for healthcare workers. With the support of conservation of resources theory, the current study aimed to investigate the association between perceived risk of COVID-19 and turnover intentions among healthcare workers, particularly Doctors, nurses, and paramedics staff. This study also aimed to investigate the mediating role of perceived fear of COVID-19 between perceived risk of COVID-19 and turnover intention. The current study also aimed to examine the buffering role that perceptions of hospital measures against COVID-19 could have on diminishing workers’ turnover intentions. Data were collected through a three time-lag email survey of healthcare workers in Pakistan (N = 178) who currently provide treatment to COVID-19 patients. The results supported the hypothesis that perceived risk of COVID-19 enhances fear of COVID-19 among healthcare workers and, consequently, their turnover intentions. Perceptions of hospital measures against COVID-19 weaken the relationship between perceived risk of COVID-19 and fear of COVID-19, which reduces turnover intentions of health care workers. The current study offers implications for theory, practitioners, and society.

## 1. Introduction

Infectious diseases are a major threat to the health and wellbeing of healthcare workers, which include doctors, nurses, and paramedics who are directly dealing with COVID-19 patients as they are at higher risk of infection due to frequent interaction with affected patients [1,2,3]. According to Chan-Yeung [4], almost one-fifth of the victims during the severe acute respiratory syndrome (SARS) outbreak were healthcare workers, including doctors, nurses, and paramedics staff. Ferioli et al. [5] also reported that 50% of healthcare workers, including doctors, nurses, and paramedics, were affected by a lethal disease before SARS. More recently, the contagious nature of COVID-19 has put healthcare workers at immense risk of contracting the virus as they have to perform their duties for infected patients [6]. Despite the growing number of studies on the spread of fear of COVID-19 among common citizens, research is still scarce on the perceived risk of COVID-19 and its negative consequences for healthcare workers, including doctors, nurses, and paramedics staff [7]. More specifically, there does not seem to be a valid scale to measure the perceived risk of COVID-19 correctly. Current research on COVID-19 is predominately dedicated to preparing a vaccine, treatment rate, and infection control [8]. Although organizational behavior researchers have started investigating the implications of COVID-19 for organizations [9,10], its psychosocial aspects [11] and its consequences for turnover intentions, particularly for healthcare workers, are less addressed [12,13]. Therefore, the current study is among the few studies to conceptualize and empirically test the perceived risk of COVID-19 by revising an existing measure available on perceived vulnerability to disease developed by Duncan et al. [14]. In the current study, we define perceived risk as a negative perception in which the individual believes it is highly likely to contract a COVID-19 infection from the surroundings.

Several studies have highlighted an increase in psychological issues [11], particularly fear due to COVID-19 among the general population [15]; however, the existing body of research is less clear about the causes for fear in doctors, nurses and paramedics staff. Fear of COVID-19 can be defined as an unpleasant emotion in which the individual feels scared of contracting the virus [16]. The limited knowledge of occupationally acquired infectious diseases suggests that healthcare workers constantly fear becoming infected [17]. This perception of fear could eventually affect their career trajectories [18].

Indeed, increased intentions of leaving have been reported among healthcare workers assigned to treating patients suffering from infectious diseases [18]. Turnover intention refers to individuals’ cognitive decision-making toward quitting the job and is considered a proxy predictor of actual voluntary turnover [19]. Some researchers have also highlighted the causes of turnover among healthcare workers. For instance, Brossoit et al. [20] studied healthcare workers’ schedule control (including sleep) and job satisfaction as the antecedents of turnover. Similarly, Li et al. [21] tested the impact of work practice environments, job engagement, and pressure on the turnover intention of nurses. Although these studies have enhanced our understanding of antecedents of healthcare worker turnover intention, little is known about the role of infectious disease in this regard. In addition to the research conducted on fear, emotional exhaustion, and job dissatisfaction faced by healthcare workers [18], the catalysts of the turnover need to be explicitly discussed, particularly in the context of infectious disease.

The lack of available treatment for COVID-19 could be a concrete cause of healthcare workers’ turnover. Employee perceptions about the measures taken against the disease, act as a deciding factor for quitting the job [22]. Existing studies have shown that healthcare workers are more likely to stay and provide treatment for infectious diseases if they perceive that the hospital has taken sufficient measures to ensure their protection [22]. COVID-19, a highly contagious disease, has also raised concerns among healthcare workers regarding the measures taken by hospitals against the infection [23]. These healthcare workers act as frontline employees whenever any pandemic hits the world [24]. Hospitals cannot afford to lose frontline staff when patients’ lives are at stake [25]. Therefore, there is an urgent need to study the role of healthcare perceptions regarding hospital measures against infection in times of worldwide health emergency [26].

Thus, a crucial step to reduce the negative consequences of health workers’ perceptions (of risk and fear) could be hospitals taking urgent measures [27]. Health workers might have positive feelings towards such measures, which could buffer risk perception and turnover intentions. Existing research on occupationally transmitted diseases shows that when health care workers witness preventive measures taken by hospitals, they are less afraid of the risk of infection [23,27]. Conversely, dissatisfaction with hospital measures against infectious disease is a proven primary reason for stress, dissatisfaction, and fear among healthcare workers [28].

To summarize, the current study has several aims. The first aim of the current study is to define and measure the perceived risk of COVID-19 by adapting an existing scale on the vulnerability of infection developed by Duncan et al. [14]. The second aim of this study is to investigate the mediating role of perceived fear of COVID-19 between perceived risk of COVID-19 and turnover intention. Lastly, the third aim of the current study is to test the role of perception of hospital measures against COVID-19 as a potential buffer for reducing the fear of healthcare workers performing their duties with the COVID-19 patients.

## 2. Theory and Hypotheses Development

### 2.1. Supporting Theory

The conservation of resources theory [29,30] has been used as an overarching theory to support the proposed model. This theory talks about how stressors and exposure to stressors threaten and depletes employee resources and energy reservoirs of employees, which they are trying to build and preserve. This theory explains resources as “objects, personal characteristics, conditions or energies valued by the individual” and suggests that employees acquire, maintain and protect such resources to meet work demands and challenges [29, p. 516]. This theory states that exposure to stressful situations depletes these resources, which ultimately affect individuals’ overall work attitude, behaviors, and mental health [31]. This theory also posits that resource gain helps in offsetting the threat to resource loss. COR theory is based on the notion that employees usually have a shortage of physical, mental, and emotional resources; they start managing their remaining resources to avoid future depletion in the event of resource loss. COR also claims that resource loss is more salient than resource gain; therefore, when employees face resource depletion from challenging job demands, they feel less energy to fight back. They utilize their existing resources to stop further resource loss [29,30].

In this study, the perceived risk of COVID-19 is a stressor and considered a threat to healthcare workers’ physical, psychological, and emotional resources. The threat of losing these resources causes fear of COVID-19 in healthcare workers. The fear of COVID-19 continues to deplete employees’ emotional, intellectual, physical, and psychological resources, and they are prone to turnover intentions to avoid further resource loss. The current study proposes the perception of hospital measures against COVID-19 as a valuable instrumental resource that helps employees to overcome the loss of resources by reducing their fear of COVID-19 in response to the perceived risk of COVID-19. Keeping in view the COR framework, we believe that employees start safeguarding their existing resources by envisioning quitting the job as a coping mechanism. COR theory also addresses the investment of existing resources to counterbalance resource loss. Relying on COR assumptions, the current research denotes the perceived risk of COVID-19 as a stressor and threat to healthcare workers’ physical, psychological, and emotional resources.

### 2.2. Perceived Risk of COVID-19 and Turnover Intentions of Health Care Workers

Infection disease research indicates that nurses in this sector are more likely to experience fear of occupationally acquired diseases, post-traumatic stress, and low job satisfaction [18]. Moreover, Wu et al. [32] imply that healthcare workers face burnout due to tough working conditions and high-risk perceptions. The existing studies also suggest that turnover intention is common among those healthcare professionals providing treatment against infectious diseases [18]. They may think about quitting their jobs to avoid further loss of resources. The turnover intention could then be considered an employee defense strategy [30]. Previous studies have also shown that nurses tend to think about quitting their job to protect their remaining resources due to job pressure [21]. Based on the assumptions of COR that stressors cause threats of resource depletion, and excessive depletion forces employees into a defensive mode in which they use every tactic to protect further resource loss, the perceived risk of COVID-19 (stressor) depletes employees’ mental efforts and energy (psychological resources) causing them to fear COVID-19 [31]. Under defensive mode, this fear prompts employees to start thinking about quitting the organization to protect their resources from depletion [33]. Hence, the current study proposes:

**Hypothesis** **1** **(H1).***Perceived risk of COVID-19 increases turnover intentions among health care workers*.

### 2.3. Mediating Role of Fear of COVID-19

In addition to the effects of being infected, perceptions of such a risk spread a wave of stress, anxiety, and fear among healthcare workers [27]. Existing literature reveals that healthcare workers report their intentions to leave work due to the risk of catching the infection [18]. The current study proposes that fear of COVID-19 mediates the relationship between the perceived risk of COVID-19 and turnover intention in a healthcare environment. COR theory [29,30] supports this notion. It focuses on resource loss, suggesting that the threat of loss of employee resources or actual loss of resources causes stress. This stress spiral gains momentum with time and, ultimately, leads to negative outcomes. We believe that the perceived risk of COVID-19 is a workplace stressor that depletes employees’ psychological, emotional, and mental resources by enhancing fear of COVID-19. As a result, healthcare workers tend to become defensive to protect their remaining energy. The thought of quitting is a defense mechanism. The current study, therefore, proposes:

**Hypothesis** **2** **(H2).**
*Fear of COVID-19 mediates the relationship between the perceived risk of COVID-19 and turnover intention among healthcare workers.*


### 2.4. Moderating Role of Perception of Hospital Measures against COVID-19

COR theory also focuses on resource investment, which states that people can minimize resource loss by investing in resources available to them [30]. The current study posits the perception of hospital measures against COVID-19 as an instrumental resource to help employees overcome resource loss by mitigating their fear of COVID-19 [34]. Studies have found that healthcare workers who believe that their hospital has taken sufficient measures against COVID-19 are experiencing less fear despite the risk [23]. Multiple studies have shown that those healthcare workers are less likely to develop turnover intention who receives organizational support [35,36]. The literature on healthcare workers indicates that lack of perceived organizational support is among the several factors that enhance turnover intention and other negative behaviors [37,38]. Previous studies also claim that fear caused by the contagious disease can be reduced when healthcare workers are confident that hospital management prioritizes their safety [22]. In short, the perception of hospital measures against COVID-19 can be an important resource to help employees safeguard their resources. Therefore, the current study proposes:

**Hypothesis** **3** **(H3).**
*Perceptions of hospital measures against COVID-19 moderate the relationship between perceived risk and fear of COVID-19 so that a positive relationship between the two will be weaker in cases of a high level of perception of hospital measures against COVID-19.*


Figure 1 shows the proposed hypothesized model.

## 3. Methods

### 3.1. Participants and Procedure

The current study adheres to the STROBE checklist for reporting quantitative studies. We verify that all the essential elements of the STROBE reporting method have been added to the manuscript. Ethical approval was taken from the relevant ethics approval committee (ethical approval no. is REAC: 2020/12), and informed consent was taken from the respondents for data collection. To test the hypotheses, data were collected through a questionnaire survey in three temporally segregated time shots with a seven-day gap to reduce common method bias [39]. The data collection process was started on 15 April 2020, and ended on 14 May 2020. Pakistan confirmed its first two cases of COVID-19 on, 26 February 2020 [40]. On 23 March, a spokesperson who was representing Doctor’s union said, “We do not have personal protective equipment (PPE), or goggles, and even [face] masks we are buying from our funds” [41]. On 31 March, 26 deaths were confirmed, 1865 cases were reported [42]. This number increased exponentially in the coming days, with the figure crossing 4000 on 7 April 2020, whereas these cases reached 10,000 on 22 April 2020 [42]. During the data collection period, the pandemic took the form of an outbreak and spread quickly across the country. According to the data shared by National Emergency Operation Centre, 253 healthcare workers of Pakistan have been diagnosed with COVID-19 so far, mainly because of a lack of proper protective equipment. This number might increase [43]. The study subjects were healthcare workers (doctors, nurses, and paramedic staff) currently treating COVID-19 patients in different Pakistani hospitals.

Respondents were selected using a non-probability snowball sampling technique because it was difficult to approach healthcare workers treating COVID-19 patients. On the recommendations of Penrod et al. [44], extra efforts were made to increase the diversity of the snowball sample to increase its generalizability. For this purpose, the authors chose respondents from contact lists and asked them to refer additional respondents further. Kirchherr and Charles [45] argue that sample seed diversity helps overcome the drawbacks of snowball sampling. Hence, the authors randomly selected respondents from those referred for data collection to enhance researcher control and minimize bias. Participation was voluntary, and respondents were ensured that their confidentiality would be maintained. The survey instrument was mailed to the respondents after acquiring their consent.

To test the hypotheses of the study, data were collected through questionnaires in three temporally segregated time shots with a seven-day gap to reduce common method bias [39]. A total of 350 surveys were circulated at time 1 (T1) to collect data about respondents’ demographic information, perceived risk of COVID-19 and their perception of hospital measures against COVID-19; out of 350, 281 responses were received back. At time 2 (T2), the 281 respondents of time one were contacted again to report their fear of COVID-19 and only 228 responses were obtained at the end of T2. In time 3 (T3), the 228 respondents of T1 and T2 were contacted to provide data about turnover intention due to COVID-19. A total of 178 respondents filled the questionnaire at all three time lags. The response rate turned out to be 50.8%. Other time lag studies have also faced the drop out of response rate [46,47,48,49]. One of the significant reasons behind a lower response rate is the tough schedule and workload of the healthcare workers due to a continuous increase in COVID-19 patients. Hence, further analysis was conducted on the sample size of 178.

G*Power (Apponic, US, version 3.1.9.4) software for calculating power analysis was employed to check the adequacy of the collected sample size [50]. In a priori analysis, numbers of predictors were set to two with other default parameters (i.e., α level = 0.05, the medium effect size of 0.15, and high power of 0.95). As a result, the computed sample size was 107, which is less than the collected sample size of 178, supporting the adequacy of the collected data sample size. Furthermore, a post hoc power analysis was employed with effect size calculated based on R^2^ = 0.33 and other default parameters mentioned above for a sample size of 178. As a result, the power value of 0.99 was observed higher than the cutoff criteria 0.80 recommended by Cohen [51].

Among 178 respondents, 104 were female, and 74 were male. In addition, 75% of the respondents were between 21 and 40 years of age. Ninety-eight respondents had a nursing diploma or bachelor’s degree in healthcare education, 64 had a master’s or higher education, and 16 were doctors with MBSS degrees. In addition, 68% of the respondents had job experience of up to three years, and 32% had more than three years of experience. Of 178 respondents, 16 were doctors, 103 were nurses, and 59 were paramedics (see Table 1).

### 3.2. Measures

Responses for all variables were taken at a five-point Likert scale, ranging from 1 = strongly disagree and 5 = strongly agree.

Perceived risk of COVID-19 was measured with the scale developed for infectious disease by Duncan et al. [14]. The items of the scale were slightly modified by replacing “infectious disease” with “COVID-19” and deleting the four items from the original scale developed by Duncan et al. (item 9, 10, 11, and 12) that did not capture the risk of infectious disease in an organizational context. More specifically, these items presumed that people had participated in activities involving a stated object (e.g., public telephone, chewed pencil, used clothes, water bottle). For instance, one deleted item stated, “I avoid using public telephones because of the risk that I may catch something from the previous user.” The reliability, dimensionality, and validity of the remaining 11-item scale were tested through corrected item-total correlation, reliability analysis, exploratory factor analysis, and, later, confirmatory factor analysis (CFA). The inter-item corrected correlation of 11 items shows the average bond strength between the items with values ranging between 0.43 and 0.60; hence 11 items were retained (see Table 2). Table 3 shows the scale items with their descriptive statistics, including mean and standard deviation. The Cronbach alpha reliability value was 0.83, which is greater than 0.7, the cutoff criterion. (see Table 4). Our goal was to explore and confirm the uni-dimensionality construct of the perceived threat of COVID-19 in health care workers. The exploratory factor analysis results for the 11-item loadings on one factor adequately ranged from 0.54 to 0.70 (see Table 2). CFA confirmed the validity of the one-factor model in maximum likelihood estimation by allowing error term to covary for item 4 with item 6, item 5 with 7, and item 8 with 10, due to theoretical overlap of the questions to obtain better model fit results [52]. The 11-item loadings on a single factor in CFA ranged between 0.48 and 0.67.

Finally, model fitness was tested using the most frequently reported indicators on Jackson et al. [52]. The model fit indicators for the one-factor model yielded reasonable fit indices χ^2^ = 71.24, df = 0.41, *p* < 0.05, CFI = 0.93, TLI = 0.91, RMSEA = 0.06 based on the recommended cutoff values of 0.90 or above for CFI and TLI—combined with 0.08 and below for RMSEA (Hair et al., 2014). The above results proved that the 11-item uni-dimensional measure of the perceived risk of COVID-19 had a high corrected item-total correlation, high reliability, high factor loadings, and good model fitness. 

Fear of COVID-19 was measured using a seven-item scale developed by Ahorsu et al. [15]. A sample item is “I am most afraid of coronavirus 19.” The alpha reliability of the scale for this study was 0.87.

Turnover intention was measured with a three-item scale developed by Vigoda [53]. A sample item was “In the current situation, I often think about quitting.” The alpha reliability value of this scale for the current study is 0.72.

Perception of hospital measures against COVID-19 was measured by a three-item scale developed by Choi and Kim [54]. The sample item was “My hospital is equipped with facilities sufficient for preventing the spread of COVID-19,” The alpha reliability value of this scale is 0.74 for the current study.

### 3.3. Confirmatory Factor Analysis

CFA was performed to test the fitness of data to the hypothesized four-factor model. The proposed four-factor model (perceived risk of COVID-19, fear of COVID-19, turnover intentions, and perception of hospital measures against COVID-19) demonstrated good fit indexes based on the most widely reported fitness indicators χ^2^ = 340.07, df = 247, *p* < 0.05, χ^2^/df = 1.39, CFI = 0.93, TLI = 0.92, RMSEA = 0.05 [52]. The four-factor model fitness values range from the recommended cutoff value 0.90 or above for CFI and TLI and 0.08 or below for RMSEA [55]. Furthermore, χ^2^/df was also provided because it is less sensitive to small sample sizes and its fit criteria was less than 5 [56]. These model fitness results indicated that respondents provided adequate data on the survey instrument for the four-factors model.

## 4. Results

### 4.1. Correlation Analysis

Table 4 reflects the correlation, descriptive statistics, and reliability results of the variables under study. The correlation of perceived risk of COVID-19 was positive and significant with both fear of COVID-19 (*r* = 0.34, *p* < 0.01) and turnover intentions of health care workers (*r* = 0.37, *p* < 0.01). In addition, fear of COVID-19 was found in significant correlation with healthcare workers’ turnover intentions (*r* = 0.44, *p* < 0.01). The perception of hospital measures against COVID-19 had a negative correlation with the perceived risk of COVID-19 (*r* = −0.40, *p* < 0.01), fear of COVID-19 (*r* = −0.34, *p* < 0.01), and turnover intentions of health care workers (*r* = −0.27, *p* < 0.01).

### 4.2. Testing Hypotheses

Table 5 represents the results of the direct and mediation hypotheses. The Hayes PROCESS macros model 4 was utilized to test direct and indirect effects. As hypothesized, the perceived risk of COVID-19 influenced health care workers to revisit their career decisions; it had a significant impact on turnover intention (*β* = 0.39, *p* < 0.01). Hence, hypothesis 1 of the study was supported. In addition, the assumptions of the mediation hypothesis were satisfied with the significant impact of the perceived risk of COVID-19 on fear of COVID-19 (*β* = 0.42, *p* < 0.01) and the significant effect of fear of COVID-19 on turnover intentions of healthcare workers (*β* = 0.47, *p* < 0.01).

Furthermore, the indirect effect of the perceived risk of COVID-19 on turnover intentions of healthcare workers through the mediating mechanism of fear was also significant, with Bootstrap 5000 95% confidence interval results (Indirect effect = 0.20, LL = 0.11, UL = 0.31). The partially and completely standardized indirect effect of the perceived risk of COVID-19 on turnover intentions of healthcare workers were found significant (see Table 5). Moreover, the ratio of indirect effect to direct effect and the total effect of the perceived risk of COVID-19 on turnover intentions of healthcare workers were also found significant (see Table 5). Finally, the R-squared mediation effect size for the mediating role of fear of COVID-19 between the perceived risk of COVID-19 and turnover intentions was also found significant with no zero between the upper and lower limit 95% confidence intervals (Effect = 0.08, LL = 0.03, UL = 0.15). Thus, hypothesis 2 was supported as well.

Table 6 reveals the results for moderation analysis. To test the moderating effect of the perception of hospital measures against COVID-19 on the relationship between the perceived risk of COVID-19 and fear of COVID-19, model 1 of Hayes PROCESS macros was utilized. Model 1 is considered the best tool because, along with an interactive effect, it provides results of the slope test to understand the interactive effect direction better. Perception of hospital measures against COVID-19 and perceived risk of COVID-19 were mean-centered in line with the recommendation of Aiken et al. (1991) for testing moderation. The interactive effect of the perceived risk of COVID-19 and perception of hospital measures against it was negative and significant (*β* = −0.23, *p* < 0.01).

Furthermore, the R square change (∆R^2^ = 0.02, *p* < 0.05) was significant for the interactive effect. Slope test results justify that the relationship between the perceived risk of COVID-19 and fear of COVID-19 was weaker when employee perceptions of hospital measures against COVID-19 were higher. Hence, hypothesis 3 of the study was also supported. The moderation graph is also presented in Figure 2.

## 5. Discussion

We are living in the midst of what has been called “the worst pandemic of the century” [57]. With the death toll reaching five figures worldwide, the world has gone into quarantine. However, healthcare workers are still performing their job duties [58]. It is frightening for them as they are at a higher risk of being infected with COVID-19 due to their continuous exposure to patients [59]. Although many studies are being conducted on COVID-19 and its consequences for the general public, we lack empirical research on the risks faced by healthcare workers and, as a consequence, their intention of leaving the profession [60]. The media reports that many doctors, nurses, and other staff have been diagnosed, and many died due to COVID-19 [26,61]. Such infections and deaths have raised serious concerns about the safety of healthcare workers who work closely with infected patients [61]. In addition, it is commonly reported that many hospitals have failed to provide necessary safety equipment to healthcare providers [59]. This worsening situation increases the risk of COVID-19 and negative emotions like stress and fear among healthcare workers [60], which compels them to leave their jobs to reduce the risk of COVID-19 [59]. The current study results show that positive perception of healthcare workers about hospital measures taken against COVID-19 can help in reducing negative emotional states, such as fear of COVID-19.

The purpose of the current study was to introduce and measure the perceived risk of COVID-19 and study turnover intention as its outcome. This study also aimed to investigate the mediating role of perceived fear of COVID-19 between perceived risk of COVID-19 and turnover intention. Further, it aimed to investigate the moderating role of perceptions of hospital measures against COVID-19 between perceived risk of COVID-19 and perceived fear of COVID-19. The results of our study support our proposition that the perceived risk of COVID-19 among healthcare workers increases their level of fear of COVID-19, resulting in their turnover intentions. These results are consistent with earlier findings, which reported that healthcare workers working with the infected patients are more likely to experience fear of contracting the disease and have higher chances of quitting their job [26,59,62]. For instance, a recent study has pointed out that due to the lack of protective equipment, healthcare workers are not only on the verge of quitting their jobs but also losing their lives [59]. The current study also proposed that the perception of hospital measures against COVID-19 can helps in reducing the fear of COVID-19 among healthcare workers. In line with existing literature, the results of the current study proved that those healthcare workers who had a positive perception regarding hospital measures against COVID-19 experience a lower level of fear of COVID-19 in response to the risk of COVID-19 and these preventive measures also result in lower level turnover intention in healthcare workers [59].

These results are consistent with the conservation of resources theory as this theory states that stressors cause resource depletion, which has negative consequences. The results also showed that the perceived risk of COVID-19 which acts as a stressor consumes employee psychological resources and develops fear of COVID-19, resulting in a negative outcome that increases turnover intention. This theory also states that individuals utilize their existing resources to avoid further resource loss. The moderation results supported this notion by confirming that perception of hospital measures against COVID-19 which is a positive resource; helps to reduce resource loss by decreasing fear of COVID-19 caused as a result of the perceived risk of COVID-19. Hence, this study validates the conservation of resources theory by providing empirical support for the resource loss mechanism that comes into the action as a result of exposure to the stressors and the role of existing positive resources in coping with the negative consequences of stressors.

The current study has several contributions. First, it has defined the perceived risk of COVID-19 and adapted a scale to measure it by modifying an existing scale on disease vulnerability. This has opened doors for research on the perceived risk of infection in general and the perceived risk of COVID-19 in specific. Another contribution of this study is that it has tested the mediating and moderating mechanisms that exist between perceived risk of COVID-19 and turnover intention by proposing perceived fear of COVID-19 as a potential mediator and perceptions of hospital measures against COVID-19 as a potential moderator that reduces the impact of the perceived risk of COVID-19 on a perceived fear of COVID-19. This study contributes to the existing knowledge on infection and its detrimental effect on healthcare workers and offers several practical implications for the hospitals. The theoretical and practical implications are discussed below:

### 5.1. Theoretical Implications

This study is timely as it adds to the existing body of knowledge in several ways. First, this study has developed a new scale for measuring the perceived risk of COVID-19 by revising an existing scale on perceived vulnerability to disease [14]. Testing the psychometric properties of the latter scale was a significant contribution to the theory measuring the risk of COVID-19. Second, our study is among the limited studies highlighting the psychological and behavioral outcomes of COVID-19, particularly for healthcare workers [63,64]. Third, the current study investigated the antecedents and consequences of fear of COVID-19 for healthcare departments. In line with previous studies on healthcare workers [20,21], COR was a useful framework to explain the consequences of COVID-19 on healthcare workers’ turnover intentions. Fourth, in line with the resource gain assumption [30], we found a buffering role between healthcare workers’ perceptions of hospital measures against COVID-19 and minimized fear of COVID-19.

### 5.2. Limitations and Future Research Directions

Although the hypotheses were confirmed, the current study has some limitations. The nature of variables has restricted the present study to single-source data. Future studies could find ways to develop and test models in the context of COVID-19 through multi-source data. Although leaving one’s job could be seen as a personal choice due to loss of resources, previous research has proven the additional value of peer support to reduce turnover intention [65]. Therefore, dyadic peer responses could add value to future research.

Moreover, actual healthcare workers’ intentions in health crises could be investigated in future studies for a more accurate picture of the situation. Another limitation of the current study is that it collected data from doctors, nurses and paramedics. As they belong to three different occupational groups with different working conditions, they may react differently to workplace stressors. Hence, it is important for future researchers to conduct a comparative study to compare the differences in the turnover intention faced by doctors, nurses, and paramedics due to the perceived risk of COVID-19.

Unlike other professional roles, which can have the freedom to work virtually, healthcare workers cannot work virtually due to the nature of their job. This may cause resentment among them due to their limited career options in terms of working from home, which might cause an increase in turnover intention among them. The limitation of the current study is that it has not taken this factor into account; future researchers may consider it while replicating this study. Another limitation of our study is that we have neither directly investigated nor controlled the impact of organizational support on the turnover intention of healthcare workers. The existing literature suggests that perceived organizational support reduces turnover intention among healthcare workers [37,38]. Future researchers may also control the impact of organizational support on turnover intention to fully capture the impact of the perceived risk of COVID-19 on the turnover intention of healthcare workers. Future researchers may also control the impact of the salary of healthcare workers on their turnover intention, as the difference in the salary of doctors, nurses, and paramedics might also be a potential factor in effecting turnover intention. The perceived risk of bringing COVID-19 from work to home and also from home to work might cause increased stress levels; however, it has not been investigated in the current study. Future researchers may investigate the emotional, psychological, and behavioral outcomes of the perceived risk of bringing COVID-19 from work to home and the perceived risk of bringing COVID-19 from home to work as well.

The current study has only identified two antecedents of turnover intention: perceived risk of COVID-19 and fear of the disease. Future researchers might test other precursors, such as inadequate hospital measures against COVID-19 and work overload. Previous research among nurses has proven a positive relationship between turnover intention and inadequate pay, inequality at work, too much work, staff shortages, lack of promotions, job insecurity, and lack of management support [66]. Future studies might investigate other factors that predict turnover intention among healthcare workers.

### 5.3. Conclusion

COVID-19 is here to stay for a while as it will still take a lot of time to vaccinate everyone. Right now, our only hope lies in the healthcare staff working day and night to take care of patients despite the risk of catching infection. Hospitals should go extra mile just to ensure that healthcare workers develop positive perceptions about the measures taken against COVID-19 as it will reduce fear of COVID-19 in reaction to perceived risk of COVID-19, which will eventually lead to a decrease in their turnover intention.

### 5.4. Relevance for Clinical Practice

The results of the current study offer several practical implications to hospital managers. These results indicate that the perceived risk of COVID-19 is one of the possible causes behind an increase in fear of COVID-19 and turnover intention among healthcare workers. To reduce turnover intention, the hospital managers must take solid measures to reduce the perceived risk of COVID-19 and fear of COVID-19 among healthcare workers. The results also indicate the importance of the perception of healthcare workers regarding hospital measures against COVID-19. The hospital managers should develop an action plan to enhance positive perceptions about the hospital measures and reduce negative perceptions about the measures taken against COVID-19.

## Figures and Tables

**Figure 1 ijerph-18-10705-f001:**
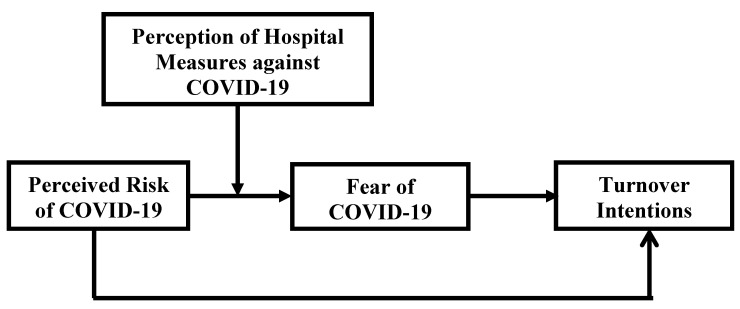
Proposed hypothesized model.

**Figure 2 ijerph-18-10705-f002:**
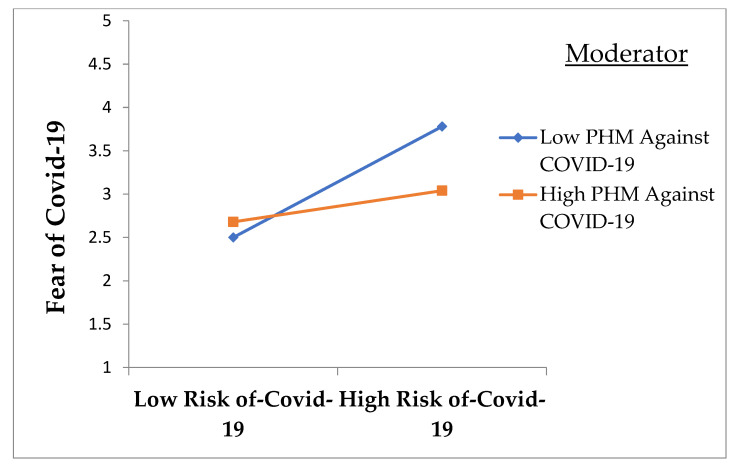
Moderation graph for the buffering effect of perception of hospital measures on the Relationship of perceived risk of COVID-19 and fear COVID-19 of healthcare workers; PHM = Perception of hospital measures.

**Table 1 ijerph-18-10705-t001:** Respondents characteristics.

Variable	Frequency	Percentage
**Gender**		
Male	74	41.6
Female	104	58.4
**Age**		
21–30 years	63	35.4
31–40 years	71	39.9
41–50 years	27	15.2
50 and Above	17	9.6
**Education**		
Nursing Diploma/ Bachelor	98	55
Master and above	64	36
MBBS	16	09
**Experience**		
Less than 1 year	51	28.7
1-3 years	70	39.3
3-5 years	32	18
5 and above	25	14
**Job Title**		
Doctor	16	09
Nurses	103	57.9
Para medics	59	33.1

Note: N = 178.

**Table 2 ijerph-18-10705-t002:** Perceived threat of COVID-19 infection scale.

S.No	Items	Corrected Item-Total Correlation	EFA Loadings	CFA Loadings
1.	In general, I am very susceptible to COVID-19.	0.55	0.65	0.60
2.	I am unlikely to catch a COVID-19, even if it is ‘going around’. (R)	0.48	0.59	0.53
3.	If COVID-19 is ‘going around’, I will get it.	0.49	0.59	0.53
4.	My immune system protects me from COVID-19 and other illnesses that other people get. (R)	0.52	0.62	0.56
5.	I am more likely than the people around me to catch COVID-19.	0.51	0.62	0.57
6.	My past experiences make me believe I am not likely to get sick due to COVID-19 even when my friends are sick. (R)	0.43	0.54	0.48
7.	I have a history of susceptibility to infectious disease like COVID-19.	0.52	0.63	0.58
8.	Due to COVID-19, I prefer to wash my hands pretty soon after shaking someone’s hand.	0.60	0.70	0.67
9.	In this time of COVID-19, It really bothers me when people sneeze without covering their mouths.	0.50	0.61	0.56
10.	In this time of COVID-19, It does not make me anxious to be around sick people. (R)	0.47	0.58	0.53
11.	In this time of COVID-19, My hands do not feel dirty after touching money. (R)	0.47	0.58	0.52

N = 178, M = Mean, S.No = Serial number. EFA = Exploratory factor analysis (extraction method: principal component analysis). CFA = Confirmatory factor analysis. Items were rated on five-point Likert scale range 1 = strongly disagree to 5 = strongly agree. (R) Indicate reverse questions. Source: modified version of perceived vulnerability to infection scale of Duncan et al. [14].

**Table 3 ijerph-18-10705-t003:** Scale items and descriptive statistics.

S.No	Items	Mean	SD
1.	In general, I am very susceptible to COVID-19.	3.94	0.80
2.	I am unlikely to catch a COVID-19, even if it is ‘going around’. (R)	3.77	0.75
3.	If COVID-19 is ‘going around’, I will get it.	3.71	0.85
4.	My immune system protects me from COVID-19 and other illnesses that other people get. (R)	3.78	0.86
5.	I am more likely than the people around me to catch COVID-19.	3.64	0.88
6.	My past experiences make me believe I am not likely to get sick due to COVID-19 even when my friends are sick. (R)	3.50	0.79
7.	I have a history of susceptibility to infectious disease like COVID-19.	3.43	0.86
8.	Due to COVID-19, I prefer to wash my hands pretty soon after shaking someone’s hand.	3.62	0.89
9.	In this time of COVID-19, It really bothers me when people sneeze without covering their mouths.	3.58	0.86
10.	In this time of COVID-19, It does not make me anxious to be around sick people. (R)	3.54	0.67
11.	In this time of COVID-19, My hands do not feel dirty after touching money. (R)	3.32	0.74

N = 178, S.No = Serial number, M = Mean, SD = Standard deviation. Items were rated on five point Likert scale range 1 = strongly disagree to 5 = strongly agree. (R) Indicate reverse questions. Source: modified version of perceived vulnerability to infection scale of Duncan et al. [14].

**Table 4 ijerph-18-10705-t004:** Mean, standard deviation, reliability, and correlation.

S.No	Variables	M	SD	1	2	3	4	5	6	7	8	9
1.	Gender	-	-									
2.	Age	1.99	0.94	0.06								
3.	Education	2.27	0.61	0.11	0.04							
4.	Experience	2.17	1.0	−0.07	0.76 **	0.02						
5.	Job Title	2.24	0.60	0.23 **	0.11	0.51 **	0.03					
6.	Perceived Risk of COVID-19	3.63	0.51	−0.03	−0.05	−0.03	0.01	−0.05	(0.83)			
7.	Fear of COVID-19	3.76	0.62	−0.03	−0.01	−0.01	0.02	−0.02	0.34 **	(0.87)		
8.	Turnover Intentions	3.28	0.81	0.07	0.05	0.01	0.00	0.09	0.37 **	0.44 **	(0.72)	
9.	PHM against COVID-19	3.11	0.92	0.01	−0.08	0.06	−0.09	0.03	−0.40 **	−0.34 **	−0.27 **	(0.74)

N = 178, * *p* < 0.05, ** *p* < 0.01; Cronbach alpha are provided bold in parentheses. Abbreviation: S.No = Serial number, M = Mean, SD = Standard deviation. PHM = Perception of hospital measures.

**Table 5 ijerph-18-10705-t005:** Bootstrapping results for direct and indirect effects.

Direct Effect	Effect	S.E		t
Perceived Risk of COVID-19 → Turnover Intentions	0.39 **	0.11		3.47
Perceived Risk of COVID-19 → Fear of COVID-19	0.42 **	0.09		4.82
Fear of COVID-19 → Turnover Intentions	0.47 **	0.09		5.17
**(95% Bias Corrected Confidence Interval method)**				
**Mediator (Fear of COVID-19)**	**Effect**	**Boot S.E**	**LL 95% CI**	**UL 95% CI**
Indirect effect of Perceived Risk of COVID-19 on Turnover Intentions	0.20	0.05	0.11	0.31
Partially standardized Indirect effect of Perceived Risk of COVID-19 on Turnover Intentions	0.24	0.06	0.13	0.39
Completely standardized Indirect effect of Perceived Risk of COVID-19 on Turnover Intentions	0.12	0.03	0.06	0.19
Ratio of indirect to total effect of Perceived Risk of COVID-19 on Turnover Intentions	0.34	0.10	0.18	0.58
Ratio of indirect to direct effect of Perceived Risk of COVID-19 on Turnover Intentions	0.51	0.34	0.22	1.41
R-squared mediation effect size	0.08	0.02	0.03	0.15

N = 178, * *p* < 0.05, ** *p* < 0.01. Abbreviations: LL = Lower limit, UL = Upper limit, CI = Confidence interval, S.E = Standard error.

**Table 6 ijerph-18-10705-t006:** Moderation analysis.

Moderator: Perception of Hospital Measures against COVID-19				
		Β	S.E	∆R²
Constant		3.00		
Perceived Risk of COVID-19 → Fear of COVID-19		0.41 **	0.10	
Perception of Hospital Measures against COVID-19 → Fear of COVID-19		−0.14 **	0.05	
Perceived Risk of COVID-19 x Perception of Hospital Measures against COVID-19 → Fear of COVID-19		−0.23 **	0.10	0.02 **
Conditional Effects of Moderator at M ± 1 SD (Slope Test)	**Effect**	**S.E**	**LL 95% CI**	**UL 95% CI**
PHM Low −1 SD (−0.92)	0.63	0.17	0.30	0.95
PHM Medium M (0.00)	0.41	0.10	0.21	0.61
PHM High +1 SD (0.92)	0.20	0.10	0.01	0.40

N = 178, * *p* < 0.05, *p* ** < 0.01, Abbreviations: PHM = Perception of Hospital Measures against COVID-19; LL = Lower Limit, UL = Upper limit, CI = Confidence interval, SD = Standard deviation, M = Mean, S.E = Standard error.

## Data Availability

The data that support the findings of this study are available from the corresponding author upon reasonable request.

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
