# Peer review of "The Interactive Effect of COVID-19 Risk and Hospital Measures on Turnover Intentions of Healthcare Workers: A Time-Lagged Study"

_ijerph, 2021, doi:10.3390/ijerph182010705_

Round 1

Reviewer 1 Report

Typo in Abstract line 15. Should be "enhances"

I am wondering if you might want to consider:

  • how in healthcare work has to be in-person; in other professional roles, it might be virtual; I am wondering if there is any resentment based on what might be career options and thus lead to greater turnover intent
  • what is the level of Perceived Organizational Support (check out that construct in the literature); some hospitals have been quite supportive by bringing in EAP (I have done this as a consultant) and whether POS would make a difference in turnover intentions
  • would the level of pay within the healthcare hierarchy make a difference (e.g., nurses and techs as opposed to doctors)
  • there is also a fear of healthcare workers bringing COVID from home to work and vice versa; that raises stress levels.

Author Response

REVIEWER 1

We are extremely thankful to you for taking out time to review our manuscript. We have incorporated all the changes which you suggested. Your valuable suggestions have helped us in enhancing the quality of our manuscript up to a great extent. We have addressed all your concerns. The changes suggested by you are highlighted in PINK color.

Typo in Abstract line 15. Should be “enhances”

Thanks for highlighting this mistake. We have now fixed it.

I am wondering if you might want to consider.

How in healthcare work has to be in-person; in other professional roles, it might be virtual; I am wondering if there is any resentment based on what might be career options and thus lead to  greater turnover intent

Thanks for identifying an important point. We also agree with your point of view. During COVID-19, majority of the people enjoyed the freedom of working from home but healthcare workers had to work in the hospitals. This might has caused resentment among them causing turnover intention. Though, we did not take this into the account while conducting this study but we have now mentioned it as out limitation. The paragraph is written on page 14 and is pasted below as well for your ease:

Unlike other professional roles, which can get the freedom to work virtually, healthcare workers cannot work virtually due to the nature of their job. This may cause resentment among them due to their limited career options in terms of working from home, which might cause an increase in turnover intention among them. The limitation of the current study is that it has not taken this factor into account; future researchers may consider it while replicating this study.

What is the level of Perceived Organizational Support (check out that construct in literature); some hospitals have been quite supportive by bringing in EAP (I have done this as a consultant) and whether POS would make a difference in turnover intentions

Thanks for bringing our attention to this point. We have reviewed the literature on organizational support and found sufficient evidence regarding its negative association with turnover intention. We have now cited a few studies and talked about the organizational support briefly while talking about the moderating role of perception of hospital measures against COVID-19. The literature is added on page 4 and is pasted below:

Multiple studies have shown that those healthcare workers are less likely to develop turnover intention who receives organizational support (Islam & Ahmed, 2018; Saadeh & Suifan, 2020). The literature on healthcare workers indicates that lack of perceived organizational support is among the several factors that enhance turnover intention and other negative behaviors (Albalawi, Naugton, Elayan, & Sleimi, 2019; Labrague, MnEnroe, Leo-cadio, Van Bogaert, & Tsaras, 2018).

 Additionally, we have added it as the limitation of our study that we have neither directly investigated nor controlled its impact on turnover intention on page 14. The explanation is also pasted below for your ease:

Another limitation of our study is that we have neither directly investigated nor controlled the impact of organizational support on the turnover intention of healthcare workers. The existing literature suggests that perceived organizational support reduces turnover intention among healthcare workers (Albalawi et al., 2019; Labrague et al., 2018). Future researchers may also control the impact of organizational support on turnover intention to fully capture the impact of the perceived risk of COVID-19 on the turnover intention of healthcare workers.

Would the level of pay within the healthcare hierarchy make a difference (e.g., nurses and techs as opposed to doctors)

Thanks for your query. We believe that level of pay within the healthcare hierarchy might be one of the many reasons behind an increase in turnover intention. Salary definitely makes a difference but it does not come under the scope of our study. We have now discussed it in the future research directions. We have suggested the future researchers to consider this point while replicating this study on page 14.

Future researchers may also control the impact of the salary of healthcare workers on their turnover intention, as the difference in the salary of doctors, nurses, and paramedics might also be a potential factor in effecting turnover intention.

There is also a fear of healthcare workers bringing COVID from home to work and vice versa; that raises stress levels.

 Thanks for sharing your valuable suggestion. The current study was based on the perceived risk of COVID-19. We have defined it as “a negative perception in which the individual believes it is highly likely to contract a COVID-19 infection from the surroundings.” The surroundings can be workplace or home. We agree with you that perceived risk of bringing COVID-19 from either work to home or home to work might also be the reason behind increased stress level. Hence, we have mentioned it in the limitations and future research directions section on page 14. The sentences are pasted below:

The perceived risk of bringing COVID-19 from work to home and also home to work might cause increased stress levels; however, it has not been investigated in the current study. Future researchers may investigate the emotional, psychological and behavioral outcomes of the perceived risk of bringing COVID-19 from work to home and the perceived risk of bringing COVID-19 from home to work as well. 

Additionally, the hospitals in Pakistan instructed the healthcare workers to stay at home if they face any symptoms and get them tested for COVID-19 before coming to the hospital. Those healthcare workers who got exposed to COVID-19 at home from their family members were sent to Quarantine. They were bound to avail leave for a period of at least three weeks and show negative COVID-19 test result before rejoining their job. We have now talked about it in the limitations and future research directions section.  

Once again, we are thankful to you for your valuable feedback. We hope that you will like our revised manuscript and will consider it worthy of publication in this prestigious journal.

Reviewer 2 Report

This seems to be an important global topic that is described and addressed well by the authors.  Their analysis of the findings from the study participants is valuable.  They point out appropriate steps that can be taken to decrease turnover intention related to fear of Covid-19, the availability of PPE, and hospital leaders' concerns for safety. 

The authors clearly stated limitations in the study using a single source.  Follow-up studies might explore other healthcare settings and invite subjects from all the various healthcare disciplines.

Overall, this is a well-written and referenced document. I only noticed one edit I would recommend: line 205  add "ed" to fill so that it reads "filled".

Author Response

REVIEWER 2

This seems to be an important global topic that is described and addressed well by the authors. Their analysis of the findings from the study participants is valuable. They point out appropriate steps that can be taken to decrease turnover intention related to fear of Covid-19, the availability of PPE, and hospital leaders’ concern for safety.

The authors clearly stated limitations in the study using a single source. Follow-up studies might explore other healthcare settings and invite subjects from all the various healthcare disciplines.

Overall, thus is a well-written and referenced document. I only noticed one edit I would recommend: line 205 add “ed” to fill so that it reads “filled”.

 We are thankful to you for taking out time to review our manuscript. We are glad that you liked our manuscript. Thanks for your positive feedback. We have now fixed the issue which you highlighted.

Reviewer 3 Report

Comments

  1. In the introduction, you should emphasize which part of the health professionals you are researching. Only line 192 showed that the object of the study is doctors, nurses and paramedics)
  2. Ethical approval number not specified in the methodology - line 172
  3. The correlation coefficient shows only the average bond strength -( line - 239 )Please correct this sentence! 

Author Response

REVIEWER 3

We are extremely thankful to you for taking out time to review our manuscript. We have incorporated all the changes which you suggested. Your valuable suggestions have helped us in enhancing the quality of our manuscript up to a great extent. We have addressed all your concerns. The changes suggested by you are highlighted in BLUE color.

  1. In the Introduction, you should emphasize which part of the health professionals you are researching. Only line 192 showed that the object of the study is doctors, nurses and paramedics)

Thanks for identifying this issue. As our study revolved around doctors, nurses and paramedics, we reviewed the literature that specifically talked about these three occupational roles. However, we forgot to mention it clearly in the start of the manuscript. Keeping in view your recommendations, we have now mentioned this in the abstract and also in the text on at multiple occasions to avoid ambiguity and confusion (see page 1 and page 2).

  1. Ethical approval number not specified in the methodology –line 172

Thanks for identifying this mistake. We initially provided a separate ethical approval letter but we have now mentioned the ethical approval number in the methodology section as per your guidelines on page 5. The details are also provided below for your ease:

The ethical approval code no is REAC:2020/12.

  1. The correlation coefficient shows only the average bond strength –(line – 239) please correct this sentence!

Thanks for highlighting this technical issue. We have now rewritten the sentence on page 7, it is pasted below as well.

The inter-item corrected correlation of 11 items shows the average bond strength between the items with values ranging between 0.43 and 0.60

Once again, we are thankful to you for your valuable feedback. We hope that you will like our revised manuscript and consider it worthy of publication in this prestigious journal.

Reviewer 4 Report

Comments concerning Manuscript ID: ijerph-1391607

This a three time-lag questionnaire study where perceived risk of COVID-19 and turnover intentions among healthcare workers was investigated including mediating role of fear of COVID-19 and moderating role of perceived hospital measures against COVID-19.  The study is inspired by conservation of resources theory and 178 healthcare workers consisting of nurses, doctors and paramedics were included in the study.  The results supported the proposed hypotheses showing that perceived risk of COVID-19 increased the fear of COVID-19 and thereby participants’ turnover intentions, while perceived hospital measures against COVID-19 moderated the relationship between perceived risk of COVID-19 and fear of 22 COVID-19, reducing turnover intentions. The study focuses on a current and important topic, having relevance for healthcare workers globally. However, the paper needs to be revised since there are several issues that need to be addressed; more detailed comments can be found below.

  1. The aim of the study should be stated more clearly both in the abstract and the text
  2. The theoretical connection to COR-theory is vague as resources are not directly investigated. The results should be related more clearly to COR-theory also in the Discussion section
  3. Why a seven-day gap, why not longer period? r.175
  4. The group of healthcare workers consist of three different occupational groups with probably different working conditions. So, the question is are they comparable, e.g., are they exposed to a same extent etc. This should be discussed among the limations
  5. It is unclear if there were any dropouts, i.e., did everybody who got the questionnaire also respond
  6. Some layout issues: The size of Figure 1 and Figure 2 should be reduced. Also, the Tables 2-6 should be condensed. What does S.No stand for? Note under Table 3, Source: should indicate that this is modified version of Duncan et al, 2009.
  7. 282 less than 5 or - now less than .5?
  8. Discussion, start by summarizing and discussing your results in relation to the study aim and previous research. Try to clearly highlight the unique contribution of you study.
  9. 356-358 “The current study has highlighted important issues by proposing preventive measures to resolve the psychological issues of healthcare workers.” It is unclear what you mean here and which parts of the results you are referring?
  10. 401-402 “Hospitals should take care of healthcare workers with the same devotion and kindness as workers.” Whom are you referring to by workers?
  11. 402-404 “Thus, it is of the utmost importance to enhance the morale of healthcare workers and ensure their safety so that their ideas of leaving the job do not remain an option.” Enhance the morale? This sentence need to be rephrased.
  12. 405-420. Relevance for clinical practice – here you are discussing something that you really have not investigated in the study. This needs to be related to your study.
  13. Some language issues that need to be addressed, e.g., r.18, r.174 survey questionnaire, use either survey or questionnaire

Author Response

REVIEWER 4

This a three time-lag questionnaire study where perceived risk of COVID-19 and turnover intentions among healthcare workers was investigated including mediating role of fear of COVID-19 and moderating role of perceived hospital measures against COVID-19. The study is inspired by conversation of resources theory and 178 healthcare workers consisting of nurses, doctors and paramedics were included in the study. The results supported the proposed hypotheses showing that perceived risk of COVID-19 increased the fear of COVID-19 and the thereby participants’ turnover intentions, while perceived hospital measures against COVID-19 moderated the relationship between perceived risk of COVID-19 and fear of COVID-19, reducing turnover intentions. The study focuses on a current and important topic, having relevance for healthcare workers globally. However, the paper needs to be revised since there are several issues that need to be addressed; more detailed comments can be found below.

We are thankful to you for taking out time to review our manuscript. Your feedback has helped us in enhancing the quality of our manuscript up to a great extent. We have incorporated all the changes recommended by you. The changes made as per your suggestions are highlighted in YELLOW color in the manuscript for your ease. We have addressed all your concerns below. 

  1. The aim of the study should be stated more clearly both in the abstract and the text.

Thanks for your suggestion. We have now added the aims of the study in the abstract on page 1. The revised abstract is pasted below for your ease:

the current study aimed to investigate the association between perceived risk of COVID-19 and turnover intentions among healthcare workers, particularly doctors, nurses and paramedics staff. This study also aimed to investigate the mediating role of perceived fear of COVID-19 between perceived risk of COVID-19 and turnover intention. The current study also aimed to examine the buffering role that perceptions of hospital measures against COVID-19 could have on diminishing workers' turnover intentions.

The aim is also added in the text on page 3 and is pasted below:

To summarize, the current study has several aims. The first aim of the current study is to define and measure the perceived risk of COVID-19 by adapting an existing scale on the vulnerability of infection developed by Duncan, Schaller, & Park (2009). The second aim of this study is to investigate the mediating role of perceived fear of COVID-19 between perceived risk of COVID-19 and turnover intention. Lastly, the third aim of the current study is to test the role of perception of hospital measures against COVID-19 as a potential buffer for reducing the fear of healthcare workers performing their duties with the COVID-19 patients.

  1. The theoretical connection to COR-theory is vague as resources are not directly investigated.

Thanks for highlighting this important point. We have now rewritten the theory section on page 3 to better explain how conservation of resources theory supports our proposed model. The paragraphs are pasted below:

The conservation of resources theory (Hobfoll, 1989; Hobfoll, Halbesleben, Neveu, & Westman, 2018) has been used as an overarching theory to support the proposed model. This theory talks about how stressors and exposure to stressors threaten and depletes employee resources and energy reservoirs of employees, which they are trying to build and preserve. This theory explains resources as "objects, personal characteristics, conditions or energies valued by the individual" and suggests that employees acquire, maintain and protect such resources to meet work demands and challenges (Hobfoll 1989, p. 516)……………………………………………………………………………………………………………… In this study, the perceived risk of COVID-19 is a stressor and considered a threat to healthcare workers' physical, psychological, and emotional resources. The threat of losing these resources causes fear of COVID-19 in healthcare workers. The fear of COVID-19 continues to deplete employees' emotional, intellectual, physical and psychological resources, and they are prone to turnover intentions to avoid further resource loss. The current study proposes the perception of hospital measures against COVID-19 as a valuable instrumental resource that helps employees to overcome the loss of resources by reducing their fear of COVID-19 in response to the perceived risk of COVID-19. Keeping in view the COR framework, we believe that employees start safeguarding their existing resources by envisioning quitting the job as a coping mechanism. COR theory also addresses the investment of existing resources to counterbalance resource loss. Relying on COR assumptions, the current research denotes the perceived risk of COVID-19 as a stressor and threat to healthcare workers' physical, psychological, and emotional resources.  

The results should be related more clearly to COR-theory also is the Discussion section.

Thanks for your feedback. We have now linked COR theory with our findings in the discussion section on page 13.  The paragraph is pasted below for your ease:

These results are consistent with the conservation of resources theory as this theory states that stressors cause resource depletion, which has negative consequences. The results also showed that the perceived risk of COVID-19 which acts as a stressor consumes employee psychological resources and develops fear of COVID-19, resulting in a negative outcome that increases turnover intention. This theory also states that individuals utilize their existing resources to avoid further resource loss. The moderation results supported this notion by confirming that perception of hospital measures against COVID-19 which is a positive resource; helps to reduce resource loss by decreasing fear of COVID-19 caused as a result of the perceived risk of COVID-19. Hence, this study validates the conservation of resources theory by providing empirical support for the resource loss mechanism that comes into the action as a result of exposure to the stressors and the role of existing positive resources in coping with the negative consequences of stressors.

  1. Why a seven-day gap, why not longer period? r.175

We are thankful to you for highlighting this technical point. The reason collecting data across different time lags by keeping a gap of seven days comes from the literature that supports this approach. We have discussed it in detail as follows:

Researchers recommended a separation between the measures of the predictor and criterion variables to reduce method bias (Feldman & Lynch 1988; Podsakoff, MacKenzie & Podsakoff 2012;  Podsakoff, MacKenzie, Lee & Podsakoff, 2003).This separation may be (a) temporal (i.e., a time delay between measures is introduced), (b) proximal (i.e., the physical distance between measures is increased), or (c) psychological (i.e., a cover story is used to reduce the salience of the linkage between the predictor and criterion variables). Podsakoff et al. (2003) noted that these types of separation should reduce the respondent’s ability and/or motivation to use previous answers to fill in gaps in what is recalled, infer missing details, or answer subsequent questions. A Temporal separation (i.e., a time delay between measures is introduced) does this by allowing previously recalled information to leave short-term memory.

Evidence of the effectiveness of introducing a temporal separation between the measurement of the predictor and criterion variables comes from several studies. First, Ostroff, Kinicki and Clark (2002) compared predictor-criterion variable correlations for concurrent ratings of both variables to ratings obtained after a one-hour or one-month delay. They reported that the average correlations were 32% lower after a one month delay than they were in the concurrent condition. Second, Johnson et al. (2011, study 2) examined the effects of a three-week delay on the correlation between a predictor construct and a criterion construct. Their results indicated that the correlation between the constructs was 43% smaller after a three-week delay than it was when both were measured at the same time.

Podsakoff et al. (2012) and Podsakoff et al. (2003) recommended separation between the measures of predictor and criterion, however they did not mention any specific time for segregation. Researchers are conducting time lags studies on the recommendation of previous research with different period of times between the measures of predictors and criterions. For instance; Raja, Azeem, Haq, & Naseer, (2020) conducted their study with five weeks gap between time lags. Majeed and Fatima, (2020) use 15 days gap between time lags. Naseer, Khawaja, Qazi, Syed, and Shamim (2021) provided a gap of 1 to two weeks between different time lags of their study. Therefore, it’s desirable to conduct time lags studies irrespective of the duration of time lags.

Feldman, J. M., & Lynch, J. G. (1988). Self-generated validity and other effects of measurement on belief, attitude, intention, and behavior. Journal of applied Psychology73(3), 421.

Johnson, R. E., Rosen, C. C., & Djurdjevic, E. (2011). Assessing the impact of common method variance on higher order multidimensional constructs. Journal of Applied Psychology96(4), 744.

Majeed, M., & Fatima, T. (2020). Impact of exploitative leadership on psychological distress: A study of nurses. Journal of Nursing Management28(7), 1713-1724.

Naseer, S., Khawaja, K. F., Qazi, S., Syed, F., & Shamim, F. (2021). How and when information proactiveness leads to operational firm performance in the banking sector of Pakistan? The roles of open innovation, creative cognitive style, and climate for innovation. International Journal of Information Management56, 102260.

Ostroff, C., Kinicki, A. J., & Clark, M. A. (2002). Substantive and operational issues of response bias across levels of analysis: an example of climate-satisfaction relationships. Journal of Applied Psychology87(2), 355.

Podsakoff, P. M., MacKenzie, S. B., & Podsakoff, N. P. (2012). Sources of method bias in social science research and recommendations on how to control it. Annual review of psychology63, 539-569.

Podsakoff, P. M., MacKenzie, S. B., Lee, J. Y., & Podsakoff, N. P. (2003). Common method biases in behavioral research: a critical review of the literature and recommended remedies. Journal of applied psychology88(5), 879.

  1. The group of healthcare workers consist of three different occupational groups with probably different working conditions. So, the question is are they comparable, e.g., are they exposed to a same extent etc. This should be discussed among the limitations.

Thanks for your suggestions. We have now added this important point in the limitations on page 14. The paragraph is pasted below for your ease:

Another limitation of the current study is that it collected data from doctors, nurses and paramedics. As they belong to three different occupational groups with different working conditions, they may react differently to workplace stressors. Hence, it is important for future researchers to conduct a comparative study to compare the differences in the turnover intention faced by doctors, nurses, and paramedics due to the perceived risk of COVID-19.

  1. It is unclear if there were any dropouts, i.e., did everybody who got the questionnaire are respond.

Thanks for your query. We have now provided the details regarding the dropouts in the manuscript on page 6. The paragraph is pasted below for your ease:

To test the hypotheses of the study, data were collected through questionnaires in three temporally segregated time shots with a seven-day gap to reduce common method bias (Podsakoff et al., 2012). A total of 350 questionnaires were circulated at time 1 (T1) to collect data about respondents' demographic information, perceived risk of COVID-19 and their perception of hospital measures against COVID-19; out of 350, 281 responses were received back.  At time 2 (T2), the 281 respondents of time one were contacted again to report their fear of COVID-19 and only 228 responses were obtained at the end of T2. In time 3 (T3), the 228 respondents of T1 and T2 were contacted to provide data about their turnover intention due to COVID-19. A total of 178 respondents filled the questionnaire at all three time lags.  The response rate turned out to be 50.8%. Other time lag studies have also faced the drop out of response rate (e.g. Irshad, Bartels, Majeed, & Bashir, 2021; Majeed, Irshad, Fatima, Khan, & Hassan, 2020; Raja, Azeem, Haq, & Naseer, 2020; Sarwar, Irshad, Zhong, Sarwar, & Pasha, 2020). One of the significant reasons behind a lower response rate is the tough schedule and workload of the healthcare workers due to a continuous increase in COVID-19 patients. Hence further analysis was conducted on the sample size of 178.

  1. Some layout issues: The size of Figure 1 and Figure 2 should be reduced. Also, the tables 2-6 should be condensed. What does S.No stand for? Note under Table 3, Source: should indicate that this is modified version of Duncan et al., 2009

Thanks for highlighting these issues. We have now readjusted the tables and figures as per the format of the journal. Also, we have now provided the full word for S.NO which is serial number in the notes section under each table. We have also mentioned now that the scale is the modified version of Duncan et al., 2009.

  1. 282 less than 5 or – now less than .5?

Thanks for identifying this typo. It is less than 5, we have now fixed this mistake on page 9.

  1. Discussion, start by summarizing and discussing your results in relation to the study aim and previous research. Try to clearly highlight the unique contribution of your study.

Thanks for your suggestion. We have now rewritten the discussion section keeping in view your recommendations. We have now summarized the results and discussed them in relation to the study aims and existing literature. We have also added the contribution of our research in the discussion section. The additional paragraphs are added on page 13 and are pasted below for your ease:

The current study results show that positive perception of healthcare workers about hospital measures taken against COVID-19 can help in reducing negative emotional states such as fear of COVID-19. 

The purpose of the current study was to introduce and measure the perceived risk of COVID-19 and study turnover intention as its outcome. This study also aimed to investigate the mediating role of perceived fear of COVID-19 between perceived risk of COVID-19 and turnover intention. Further, it aimed to investigate the moderating role of perceptions of hospital measures against COVID-19 between perceived risk of COVID-19 and perceived fear of COVID-19. The results of our study support our proposition that the perceived risk of COVID-19 among healthcare workers increases their level of fear of COVID-19, resulting in their turnover intentions. These results are consistent with earlier findings, which reported that healthcare workers working with the infected patients are more likely to experience fear of getting the disease and have higher chances of quitting their job (Iacobucci, 2020; Newman, 2020; Wang et al., 2020b). For instance, a recent study has pointed out that due to the lack of protective equipment, healthcare workers are not only on the verge of quitting their jobs but also losing their lives (Newman, 2020). The current study also proposed that the perception of hospital measures against COVID-19 can helps in reducing the fear of COVID-19 among healthcare workers. In line with existing literature, the results of the current study proved that those healthcare workers who had a positive perception regarding hospital measures against COVID-19 experience a lower level of fear of COVID-19 in response to the risk of COVID-19 and these preventive measures also result in lower level turnover intention in healthcare workers (Newman, 2020).

 These results are consistent with the conservation of resources theory as this theory states that stressors cause resource depletion, which has negative consequences. The results also showed that the perceived risk of COVID-19 which acts as a stressor consumes employee psychological resources and develops fear of COVID-19, resulting in a negative outcome that increases turnover intention. This theory also states that individuals utilize their existing resources to avoid further resource loss. The moderation results supported this notion by confirming that perception of hospital measures against COVID-19 which is a positive resource; helps to reduce resource loss by decreasing fear of COVID-19 caused as a result of the perceived risk of COVID-19. Hence, this study validates the conservation of resources theory by providing empirical support for the resource loss mechanism that comes into the action as a result of exposure to the stressors and the role of existing positive resources in coping with the negative consequences of stressors.

The current study has several contributions. First, it has defined the perceived risk of COVID-19 and adapted a scale to measure it by modifying an existing scale on disease vulnerability. This has opened doors for research on the perceived risk of infection in general and the perceived risk of COVID-19 in specific. Another contribution of this study is that it has tested the mediating and moderating mechanisms that exist between perceived risk of COVID-19 and turnover intention by proposing perceived fear of COVID-19 as a potential mediator and perceptions of hospital measures against COVID-19 as a potential moderator that reduces the impact of the perceived risk of COVID-19 on a perceived fear of COVID-19. This study contributes to the existing knowledge on infection and its detrimental effect on healthcare workers and offers several practical implications for the hospitals. The theoretical and practical implications are discussed below.

  1. 356-358 “The current study has highlighted important issues by proposing preventive measures to resolve the psychological issues of healthcare workers.” It is unclear what you mean here and which parts of the results you are referring?

Thanks for your query. We wanted to say that positive perceptions among healthcare workers about the hospital measures against COVID-19 reduces psychological resource loss by reducing fear of COVID-19 among them. We have now rewritten this sentence to make it clearer .The sentence is given on Page 13 and is pasted below for your ease:

The current study results show that positive perception among healthcare workers about hospital measures taken against COVID-19 can help in reducing negative emotional states such as fear of COVID-19. 

  1. 401-402 “ Hospitals should take care of healthcare workers with the same devotion and kindness as workers.” Whom are you referring as workers?

Thanks for your query. We wanted to say that hospitals should also take care of healthcare workers just like healthcare workers take care of COVID-19 patients. However, we realized that this sentence needs to be rewritten. Hence, we have now rewritten it to make it more relevant and clear. The sentence is written on page 15 and is pasted below for your ease:

Hospitals should go extra mile just to ensure that healthcare workers develop positive perceptions about the measures taken against COVID-19 as it will reduce fear of COVID-19 in reaction to perceived risk of COVID-19, which will eventually lead to a decrease in their turnover intention.

  1. 401-404 “ Thus, it is of the utmost importance to enhance the morale of the healthcare workers and ensure their safety so that their ideas of leaving the job don no remain an option.” Enhance the morale? This sentence need to be rephrased.

Thanks for your feedback. We have now edited this sentence. The conclusion section is rewritten to make it more clear and easy to understand. It is given on page 15 and is pasted below as well:

COVID-19 is here to stay for a while as it will still take a lot of time to vaccinate every-one. Right now, our only hope lies in the healthcare staff working day and night to take care of patients despite the risk of catching infection. Hospitals should go to extra lengths just to en-sure that healthcare workers develop positive perceptions about the measures taken against COVID-19 as it is only then their fear of COVID-19 will reduce, which will eventually lead to a decrease in their turnover intention.

  1. 405-420. Relevance for clinical practice – here you are discussing something that you really have not investigated in the study. This needs to be related to you study.

Thanks for your valuable suggestion. We have now rewritten the whole section. The revised section is written in accordance with the study findings. The revised paragraph is given on page 15 and is pasted below as well:

The results of the current study offer several practical implications to hospital managers. These results indicate that the perceived risk of COVID-19 is one of the possible causes behind an increase in fear of COVID-19 and turnover intention among healthcare workers. To reduce turnover intention, the hospital managers must take solid measures to reduce the perceived risk of COVID-19 and fear of COVID-19 among healthcare workers. The results also indicate the importance of the perception of healthcare workers regarding hospital measures against COVID-19. The hospital managers should develop an action plan to enhance positive perceptions about the hospital measures and reduce negative perceptions about the measures taken against COVID-19.

  1. Some language issues that need to addressed, e.g., r.18, r.174 survey questionnaire, use either survey or questionnaire

Thanks for highlighting these issues. We have now fixed them.

Once again, we are thankful to you for your valuable feedback. We hope that you will like our revised manuscript and consider it worthy of publication in this prestigious journal.